# Preparation and Applications of Silver Nanowire-Polyurethane Flexible Sensor

**DOI:** 10.3390/s25165191

**Published:** 2025-08-21

**Authors:** Jiangyin Shan, Jianhua Qian, Ling Lin, Mengrong Wei, Jingyue Xia, Lin Fu

**Affiliations:** Textile Science and Engineering, Zhejiang Sci-Tech University, Hangzhou 310020, China; 2022327100122@mails.zstu.edu.cn (J.S.); 2022327100130@mails.zstu.edu.cn (L.L.); weimengrong2021@163.com (M.W.); 18154307082@163.com (J.X.); 202230202095@mails.zstu.edu.cn (L.F.)

**Keywords:** silver nanowires, flexible sensors, high sensitivity

## Abstract

To expand the application of silver nanowires (AgNWs) in the field of flexible sensors, this study developed a stretchable flexible sensor based on thermoplastic polyurethane (TPU). Initially, the TPU nanofiber membrane was prepared by electrospinning. Subsequently, high-aspect-ratio AgNWs were synthesized via a one-step polyol reduction method. The AgNWs with the optimal aspect ratio were selected for the conductive layer and spray-coated onto the surface of the TPU nanofiber membrane. Another layer of TPU nanofiber membrane was then laminated on top, resulting in a flexible thin-film sensor with a “sandwich” structure. Through morphological, chemical structure, and crystallinity analyses, the primary factors influencing AgNWs’ growth were investigated. Performance tests revealed that the prepared AgNWs had an average length of approximately 130 μm, a diameter of about 80 nm, and an average aspect ratio exceeding 1500, with the highest being 1921. The obtained sensor exhibited a low initial resistance (26.7 Ω), high strain range (sensing, ε = 0–150%), high sensitivity (GF, over 19.21), fast response and recovery time (112 ms), and excellent conductivity (428 S/cm). Additionally, the sensor maintained stable resistance after 3000 stretching cycles at a strain range of 0–10%. The sensor could output stable and recognizable electrical signals, demonstrating significant potential for applications in motion monitoring, human–computer interaction, and healthcare fields.

## 1. Introduction

With the current deep integration and widespread application of the Internet of Things (IoT), artificial intelligence (AI), and wearable and implantable technologies [1,2,3,4,5,6], the development of flexible sensors has garnered significant attention. Compared to conventional rigid sensors, flexible sensors, as a novel category of electronic devices, exhibit distinctive characteristics such as high resilience, excellent tensile properties, superior ductility, and light weight [7,8]. These features provide them with unique advantages in various cutting-edge fields. Yao et al. [9] developed a flexible sensor capable of conforming to human skin, and Lu et al. [10] fabricated a flexible temperature sensor using printing technology [11,12,13,14]. These studies undeniably demonstrated the immense developmental potential of flexible sensors in various domains, including smart wearable devices, electronic skin, healthcare, smart homes, and robot tactile sensing [11,12,13,14].

AgNWs [15,16], as an emerging nanomaterial, exhibit outstanding conductivity, high specific surface area, stable chemical properties, and excellent flexibility, making them an ideal choice for flexible conductive materials. YANG et al. [17] combined AgNWs with polydimethylsiloxane (PDMS) to fabricate sensors with a sandwich structure, demonstrating high stability in electrical properties. Li et al. [18] prepared a composite flexible film by combining AgNWs with hydroxypropyl cellulose (HPC), showing that the flexible sensor fabricated with AgNWs exhibits high current sensitivity and excellent cycling stability. However, AgNW-based flexible thin-film sensors have faced certain challenges in terms of integration and compatibility, necessitating further research and optimization. Therefore, it is important to select a substrate with appropriate elasticity and flexibility for flexible sensors [19,20].

In recent years, researchers have prepared various flexible substrates through multiple methods, such as electrospinning and piezoelectrics [21,22]. WANG et al. [23] fabricated a substrate based on P(VDF-TrFE) electrospun membranes, and CAI et al. [24] employed hydrogel films as the substrate for flexible sensors. However, with the progression of the era, there has been increasing demand for stability, conductivity, and strain range in flexible sensors.

Therefore, a flexible sensor with a “sandwich” structure was fabricated using a thermoplastic polyurethane (TPU) electrospun film as a substrate to enhance the adhesion of AgNWs. The morphology and crystal structure of AgNWs synthesized via the one-pot polyol method were analyzed through SEM, TEM, and XRD tests. Ultimately, AgNWs with an average aspect ratio of 1500 and a maximum aspect ratio of 1921 were selected as the conductive layer, which is much higher than most of the reported AgNWs (mostly <1000). The sensor was subjected to a series of rigorous evaluations, including tensile testing, cyclic stability testing, and other assessments. Furthermore, the electrical signal exhibited fluctuations due to its high sensitivity (GF max. 19.21), excellent conductivity (428 S/cm), and fast response time (112 ms). These properties underscore the potential applications of flexible sensors in various fields, including human–computer interaction, motion monitoring, and healthcare. Furthermore, the present study expanded the 3 × 3 array sensors, providing a novel perspective and direction.

## 2. Experimental Section

### 2.1. Materials and Chemicals

Polyvinylpyrrolidone (PVP, M_w_ = 1.3 × 10^6^, 5.4 × 10^4^), anhydrous copper chloride (CuCl_2_, 98%), and polyurethane (TPU) were supplied by Shanghai Aladdin Biochemical Technology Co., Ltd. in Shanghai, China. Ethylene glycol (EG), ethanol (C_2_H_5_OH), and N,N-dimethylacetamide (DMAC) were provided by Hangzhou Gaojing Fine Chemicals Co., Ltd. in Hangzhou, China. Silver nitrate (AgNO_3_) was obtained from Guangzhou Guanghua Technology Co., Ltd. In Hangzhou, China. All chemicals were used without further purification.

### 2.2. Preparation of TPU Membranes and Nanofiber Membranes

First, 9 g of TPU and 41 g of dimethylacetamide (DMAC) were weighed into a conical flask to prepare an 18 wt% solution. The solution was stirred in a lab water bath at 70 °C for 9 h until it became transparent and homogeneous. Then, the solution was left to stand in a lab oven at 70 °C overnight. By setting the table-top film applicator to 70 μm, a TPU film was successfully fabricated.

Next, 1.8 g of TPU and 8.2 g of DMAC were weighed into a sample bottle to prepare an 18 wt% solution. Then, 5 mL of the aforementioned spinning solution was loaded into a syringe and fixed onto the electrospinning apparatus. The spinning voltage was set to 14 kV, the distance between the flat needle tip and the collecting roller was maintained at 15 cm, and the rotation speed of the collecting roller was set to 250 rpm. The injection pump flow rate was adjusted to 0.8 mL/h, and the spinning temperature was maintained at 25 ± 2 °C. After 6 h of electrospinning, a TPU nanofiber membrane was successfully fabricated. The detailed preparation process of the TPU nanofiber membrane is illustrated in Figure 1b.

### 2.3. The Preparation of AgNWs

Here, 0.69 g of PVP (Mw: 5.8 × 104) and 1.17 g of PVP (Mw: 1.3 × 106) were separately weighed and dissolved in 50 mL of EG. The PVP powder was dissolved using a magnetic stirrer (From Shanghai Aladdin Biochemical Technology Co., Ltd. in Shanghai, China) until the solution became homogeneous and stable. The prepared solution was slowly poured into the three-necked flask, which had been rinsed with EG. Subsequently, 1 mL of 3 mmol/L CuCl_2_ (dissolved in EG) was added dropwise into the three-necked flask. A magnetic stir bar (From Hangzhou Gaojing Fine Chemicals Co., Ltd. in Hangzhou, China) was placed inside, and the flask was sealed with a glass stopper (From Hangzhou Gaojing Fine Chemicals Co., Ltd. in Hangzhou, China). The flask was then fixed in a lab oil bath at 150 °C and stirred continuously at 200 rpm for 1 h (Figure 1a).

In a completely light-free environment, 0.5 g of AgNO_3_ was weighed, and 25 mL of EG was measured as the solvent. The solution was stirred using a magnetic stirrer until it became homogeneous. The reaction solution of AgNWs and EG was dripped into a three-necked flask at 75 rpm using a peristaltic pump while maintaining a constant temperature of 150 °C. AgNWs were prepared with reaction times of 6 h, 7 h, and 8 h, and labeled, respectively, as A6, A7, and A8. Of these, A8_1_ and A8_2_ represent, respectively, the underdeveloped and successfully grown AgNWs at 8 h (Figure 1a).

The three-necked flask was opened and allowed to cool to room temperature in the fume hood. The solution was then aliquoted into centrifuge tubes. To each centrifuge tube, 1 mL of the AgNW mixed solution and 2 mL of ethanol were added, followed by centrifugation in a centrifugal machine at 8000 rpm for 20 min. The supernatant was discarded. Subsequently, 6 mL of anhydrous ethanol was added, and the centrifugation process was repeated four times to obtain purified AgNWs. The purified AgNWs were dispersed in anhydrous ethanol to prevent aggregation.

### 2.4. The Fabrication of Flexible Sensors

Two pieces of TPU film, each measuring 3.5 cm × 3.5 cm, were cut. One piece of TPU film was fixed in a Petri dish, and the ethanol-dispersed AgNWs solution was uniformly coated onto the surface of the TPU film using the drop-casting method. After the ethanol evaporated, the AgNWs formed a network structure, resulting in a transparent film. A copper sheet (thickness: 0.02 mm; length: 2 cm; width: 0.5 cm; from Hangzhou Gaojing Fine Chemicals Co., Ltd. in Hangzhou, China) was placed on the TPU-AgNWs film and secured with conductive silver paste (From Hangzhou Gaojing Fine Chemicals Co., Ltd. in Hangzhou, China). Subsequently, the other piece of TPU film was placed on top, forming a “sandwich” structure (Figure 1c) for the flexible thin-film sensor.

## 3. Results and Discussion

### 3.1. Analysis of TPU and AgNW SEM Image

In this study, two preparation methods (casting method and electrostatic spinning method) were utilized to produce TPU films, and the following is an analysis of the TPU films produced by the two different preparation methods.

SEM images of the TPU material are shown in Figure 2. As shown in Figure 2b, the TPU film fibers prepared by the electrostatic spinning method exhibit uniform thickness and a smooth surface. The micro-topography of the film shows a fibrous three-dimensional network structure, and the fiber arrangement exhibits anisotropy and the formation of multiple pores of varying sizes between the fibers, thus reducing the incidence of agglomeration phenomena. The homogeneity and stability of the TPU film prepared by the electrostatic spinning method are excellent. Figure 2c demonstrates that there is a certain scale of agglomerate structures on the surface of TPU film prepared by the casting method, and this structure is evidence that polymer crystallization partially exists on the surface of the film; it can be observed that the TPU particles appear as spheroids of varying sizes. Additionally, the agglomeration phenomenon is more serious, resulting in a rough and uneven surface on the film. The presence of pores with suitable apertures has been shown to enhance the conductivity, permeability, and impermeability of the film as well as its strength. However, heterogeneity of the thickness has been demonstrated to affect the performance of the flexible sensor. Consequently, the TPU nanofiber membrane functions better as a flexible substrate.

As the reaction time increases, the diameter of AgNWs decreases from approximately 120 nanometers at A6 to approximately 80 nanometers at A8_2_. Following the analysis of five replicate samples, a statistically significant finding emerged: the diameter of A8_2_ (~80 nm) was notably smaller than that of A7 (~100 nm) and A6 (120 nm). A visual examination of the SEM images further corroborated these findings, clearly demonstrating that the samples possessed varying concentrations. This observation indicated that A8_2_ exhibited superior suitability for utilization as a raw material for the conductive layer when compared to A6, which exhibited severe agglomeration, and A7, which exhibited sparse AgNWs. This phenomenon can be attributed to the combined effects of surface ligand-mediated growth and the Ostwald ripening process. Polyvinylpyrrolidone (PVP) plays a crucial role as a capping agent. PVP demonstrates a marked preference for adsorption onto the (100) crystal faces of Ag [25,26,27]. During the initial reaction stage (A6), the PVP concentration relative to Ag^+^ is inadequate to fully passivate the (100) faces due to a short reaction time of only 6 h. Consequently, silver atoms are deposited uncontrollably on these lateral surfaces, resulting in an increased initial diameter. As the reaction progresses to A7 (7 h) and A8_2_ (8 h), PVP gradually saturates the (100) plane. This saturation forms a steric barrier layer that effectively inhibits lateral growth. Concurrently, the (111) crystal plane, which exhibits a weak interaction with PVP, persists in facilitating axial growth. The Gibbs–Thomson effect [28] has also been observed to result in elevated surface energy in small Ag nanoparticles or irregular protrusions on the AgNW surface (more prevalent at A6 and A7). These high-energy species undergo dissolution into Ag^+^ ions, subsequently redepositing on the low-energy (111) axial surfaces of larger AgNWs in subsequent stages. This Ostwald ripening-like process serves to further refine the diameter, resulting in the more uniform and thinner AgNWs observed in A8_2_. The enhanced crystallinity of AgNWs with prolonged reaction times is attributable to several processes. In the initial stages (A6), the rapid reduction of AgNO_3_ by ethylene glycol results in a high nucleation rate, leading to the formation of numerous defects, including vacancies, dislocations, and twin boundaries. As the reaction time increased from A6 to A8_2_, the system was maintained at a stable temperature of 150 °C. This temperature is sufficient to provide activation energy for atomic diffusion. Silver atoms have been observed to migrate to defect sites, where they have been shown to fill vacancies, annihilate dislocations, and reorganize diatoms and twin boundaries into more coherent interfaces.

### 3.2. Analysis of TEM Images of AgNWs

Figure 3 shows the pattern of the two-dimensional dot composition of AgNWs. From the features of this pattern, it can be clearly seen that the AgNWs have a typical face-centered cubic structure (FCC). Figure 3a,d show that the prepared AgNWs have a thinner diameter, longer length, smoother surface, and uniform thickness, which indicates that the AgNWs prepared under the experimental optimal process are excellent. There is no agglomeration phenomenon in the reaction-prepared AgNWs, and the back layer of AgNWs can be observed by transmittance observation, which proves that the reaction-prepared AgNWs are more transparent and have better optical properties, and these good transmittance and optical properties indicate high purity to a certain extent. Figure 3c,f show the electron diffraction patterns of the AgNW samples in the composition of the two-dimensional array of dots; it can be seen that there are bright diffraction spots in the patterns, and these diffraction spots are distributed with a certain regularity, which proves that the silver nanowires have good crystallinity.

### 3.3. XRD Analysis of AgNWs

The XRD images of AgNWs are shown in Figure 3. The XRD patterns of A6 (Figure 3g), A7 (Figure 3h), and A8_2_ (Figure 3i), respectively, were tested in the range of 20°~90°, and the resulting patterns were compared with the standard XRD pattern of silver (No. #87-0597); it can be observed that the positions of the diffraction peaks in the patterns of A6, A7, and A8_2_ are consistent with those of the standard card, but the intensities of the diffraction peaks are different. As shown in Figure 3g–i, five diffraction peaks appeared at 2θ = 38.28°, 44.56°, 64.27°, 77.64°, and 81.65°; no other impurity peaks appeared, indicating that the prepared AgNWs were relatively purer. These five diffraction peaks corresponded to the FCC silver structure (111), (200), (220), (311), and (222) on the five crystal faces, which proves that the synthesized AgNWs have a face-centered cubic structure and eminent crystallinity. Furthermore, XRD results (Figure 3i) demonstrate that the (111) diffraction peak exhibits the greatest intensity at A8_2_ (intensity ≈ 4314), which is about 2.34 times that of A6 (1842), exhibiting a (111)/(200) intensity ratio of approximately 4.24, suggesting preferential growth along the (111) direction. The extended reaction time facilitates the continuous supply of Ag^+^ ions, allowing for ordered deposition along the low-energy (111) plane [29]. A thorough examination of the XRD spectra of the three samples reveals that only the intensity increased. The positions of the diffraction peaks are determined by the lattice spacing, which is a characteristic of the crystal structure of AgNWs. It is evident that, in the absence of alterations to the crystal structure, the lattice spacing will remain constant. The enhancement in peak intensity with reaction time is predominantly attributable to the elevated crystallinity. As the reaction progresses, the number of ordered crystals increases, and defects are gradually repaired. This results in an increased number of crystal planes capable of coherent diffracting X-rays, leading to an increase in peak intensity [30]. Furthermore, the enhanced orientation growth along the (111) direction in A8_2_ suggests the presence of a greater number of (111) planes conducive to crystallization with a diffraction orientation, thereby resulting in higher (111) peak intensity compared to A6 and A7 [31].

### 3.4. Visualization of the Performance of AgNW-TPU Flexible Sensors

The preparation of multiple sets of samples is imperative, with each set exhibiting a uniform coating of 3 mL of AgNWs at varying concentrations. Given the tendency of alcohol to evaporate, the thickness of the final system can be regulated by adjusting the concentration and dosage of the alcohol component. When the AgNW concentration is below 18 wt%, the resistivity undergoes a precipitous decline with increasing concentration. The phenomenon can be attributed to the fact that, at low concentrations, AgNWs are sparsely dispersed, resulting in insufficient formation of conductive pathways. However, when the concentration exceeds 18 wt%, the decrease in resistivity slows down and stabilizes. At this stage, AgNWs will have formed a continuous, dense conductive network, and further increasing the concentration has a limited effect on the optimization of the pathways (Figure 4a). When the dosage is controlled between 1 and 2 mL, the conductive network is insufficiently developed, and the sensor cannot be reused after multiple cycles of stretching, resulting in a short lifespan. As the dosage increases, the sensor resistance undergoes a slight increase; however, it demonstrates commendable durability, with a recorded number of over 3000 cycles. In order to achieve the lowest possible cost while ensuring optimal performance, this experiment selected 18 wt% AgNWs at a dosage of 3 mL as the conductive layer (Figure 4b). Following the implementation of the SEM measurement of the thickness of the AgNWS conductive layer, it was determined that the optimal thickness falls between 2.3 and 3.5 μm.

Conductivity and response time are critical performance indicators for sensors. A comparative analysis of the conductivity and response time of diverse flexible conductive materials reveals that the A8_2_ sample in this study exhibited a conductivity of 428 S/cm [32,33,34,35,36,37] and a minimum response time of 112 ms [38,39,40,41,42], thereby demonstrating a substantial superiority over other composite materials and fabrics depicted in the figure. Consequently, the conductivity and response time of AgNW-TPU sensors further substantiate their potential value in future markets.

### 3.5. Performance Analysis of Flexible Sensors

As shown in Figure 5f, the stress–strain curve presents a linear relationship within 0–100% strain. This modulus reflects the sensor’s stiffness in the elastic phase, enabling it to resist excessive deformation under external forces while maintaining recoverability—upon removal of the force, the sensor regains its original form without permanent deformation, which underpins its fast response/recovery time (112 ms) and stable cyclic performance (3000 cycles). When strain increases to 100–400%, the curve deviates from linearity, indicating the onset of plastic deformation characterized by irreversible structural damage, such as the partial fracture of AgNW networks and permanent slippage of TPU molecular chains. At approximately 500% strain, the stress plummets to zero, signifying complete fracture of the sensor material.

AgNW-TPU sensors’ resistance response to tensile strain (0% to 150%) is due to the evolution of their series–parallel structure. The correlation between this structure and substrate deformation directly affects the sensors’ sensitivity.

In the 0% to 17% strain range, the TPU substrate barely deforms, and the AgNWs stay densely overlapping (Figure 5a). This creates a dominant multi-path parallel structure with many cross-contact points. At this point, the contact resistance is low, and the multi-path redundancy stabilizes the overall resistance at 26.7 Ω. Slight stretching slightly increases the spacing between some contact points, but the parallel dominant effect causes ΔR/R_0_ to grow slowly: GF = 2.44. Additionally, the uniform distribution of contact points ensures a high linear fitting degree (R^2^ = 92.9%).

As strain increases from 17% to 65%, the oriented stretching of TPU molecular chains triggers network restructuring. Most parallel contact points slip or break away, and the remaining contact points have smaller contact areas. This leads to increased contact resistance. The unbroken AgNWs form a “chain-like series structure” along the stretching direction (Figure 5a). The intrinsic resistance of a single nanowire and the contact resistance are connected in series and superimposed, making the series resistance the dominant factor and the parallel resistance the secondary factor. This significantly increases the GF to 19.29. As the strain continues to increase, microdefects stabilize, and the linear relationship between the changes in resistance and strain strengthens. The fitting degree increases to 98.1%.

In the high strain range of 65% to 150%, TPU deformation is maximal, and AgNWs undergo local fractures due to excessive stretching. The original long series chains disintegrate into short segments, forming new series structures. A small portion of the short nanowires re-contact in the wrinkled regions, forming temporary parallel branches that partially compensate for the increase in resistance. The combined effect of these two factors reduces GF to 14.58, and the regularity of the structural reorganization achieves a linear fitting degree of 99.8%. This demonstrates the adaptability of the network structure.

Additionally, tensile–sensitivity curves of sensors prepared from batches A6 and A7 were examined. In Figure 5e–g, high-aspect-ratio AgNWs (A8_2_) exhibited a substantially higher GF (19.29) compared to those of A7 (GF ≈ 16.34) and A6 (GF ≈ 11.84). This elevated GF can be attributed to the presence of denser network connections and enhanced tensile tolerance. Additionally, it was observed that the conductive network of A8_2_ exhibited resilience, maintaining its integrity even under strain levels of up to 150%.

Response time is a critical metric of the dynamic performance of flexible sensors and is fundamentally related to the rate at which electronic transmission paths are reconfigured within the conductive network. The AgNW-TPU sensor demonstrates a response time and recovery time of 112 ms at 1% strain (Figure 5b–d). According to the RC delay theory [43], the signal transmission time constant is reduced by a low-resistance network (26.7 Ω). Furthermore, the three-dimensional porous structure of the electrospun TPU membrane provides anchoring for AgNWs. During minor deformations, the contact points of the nanowires undergo elastic displacement without breaking, enabling rapid restoration of the conductive pathway without requiring atomic diffusion or interface reconstruction.

The disparities in V-I characteristics under varying static strains (Figure 5i,j) are attributable to variations in the topological structure of the conductive network. At low strains (6–18%), the AgNW network displays a dense parallel configuration, with electron conduction primarily via direct tunneling and metal contact, resulting in a linear and stable V-I curve. At a current of 0.2 A, the voltage increase is only 6.32 V, as the redundant parallel paths average out local changes. At elevated strain levels, the network predominantly functions in a series configuration, necessitating the overcoming of elevated tunneling barriers by electrons. Furthermore, periodic slippage at contact points leads to fluctuations in tunneling resistance, resulting in significant nonlinearity in the V-I curve. The enhanced electric field effect in the high-voltage region further amplifies instability.

As demonstrated in Figure 5k,l, the resistance change stability tests validate the strain dependence of the network structure. In the range of small strains between 1% and 6%, there is an observed increase in resistance from 26.2732 Ω to 27.0810 Ω. This fluctuation is recorded to be less than 0.5% within a span of five seconds. This phenomenon can be attributed to the “elastic compression” contact between AgNWs, where alterations in area are found to be reversible, and the percolation threshold remains stable [44]. At strains ranging from 25% to 150%, the resistance exhibited exponential growth with irregular fluctuations, attributed to partial plastic deformation or fracture of the nanowires. Fractured segments reconnect via substrate wrinkles, and oxidation at contact points and atomic migration cause resistance drift [45]. However, at 150% high strain, the resistance remains macroscopically stable thanks to the elastic recovery force of TPU enabling broken nanowires to form “bridge-like connections” to maintain connectivity [46].

In summary, the sensor’s electrical response is the result of “material–structure–performance” synergy. With small strains, the redundant conduction of the parallel network provides stable linear behavior, suitable for high-precision micro-deformation detection. At large strains, the tunneling mechanism of the series network leads to a nonlinear response, and the wide measurement range characteristics can meet the requirements for monitoring violent movements, providing performance boundaries and a theoretical basis for scenario-based applications.

### 3.6. Cyclic Stability of the Sensor

The cyclic stability of the resistor is a pivotal indicator of the sensor’s performance. It exerts a direct influence on the measurement accuracy and reliability of the sensor in addition to determining its service life and application range. This section presents the findings of a study examining the dynamic resistance cycling performance of AgNW-TPU flexible sensors. To this end, the resistance changes of AgNW-TPU sensors when they are stretched at different rates under the same strain were detected. The fiber membrane was stretched at different rates (0.5, 1.0, 1.5, and 2.0 mm/s) at 50% strain, and the cyclic resistance curves were measured. As demonstrated in Figure 6a, the cyclic resistance curves at constant rates exhibited high consistency, with resistance measurements at varying rates displaying a modest increase, though this is not statistically significant. This finding suggests that the relative resistance change in AgNW-TPUs is largely unaffected by the stretching rate, thereby demonstrating the resistance response’s independence from the stretching rate. The sensing stability is crucial for the repeatability of the sensor in practical applications. During deformation processes such as stretching and bending, the resistance change directly affects the sensor output signal. Excessive or unstable resistance changes can lead to signal drift and hysteresis, consequently reducing the sensor’s repeatability. The investigation into the cyclic testing of the sensor at 0–10% tensile strain sensing, as illustrated in Figure 6b–d, is a crucial aspect of this study. The outcomes demonstrate that the sensor undergoes a resistance change within the range of 0% to 2%, following 3000 tensile cycles within the 0% to 10% strain range. The sensor’s satisfactory cyclic stability indicates its capacity to preserve consistent resistance fluctuations during repetitive tensile and recoil procedures, ensuring precise signal surveillance and delivering dependable performance across diverse application domains.

### 3.7. Monitoring of Human Health

AgNW-TPU flexible sensors demonstrate considerable potential for integration into textiles, with the embedding of sensors into cuffs or collars enabling the real-time and continuous monitoring of vital signs for health assessment. As demonstrated in Figure 7d,e, the sensor output exhibited a distinct periodic waveform with minimal noise interference, effectively differentiating between a higher pulse rate of 113 beats per minute (bpm) following exercise and 65 bpm at rest; the sensor demonstrated an error margin of less than 2% when compared to commercial pulse testers, as determined by a group of volunteers in a series of trials, thereby, substantiating the precision of its measurements. The functionality of the sensor was further extended to encompass cervical spine monitoring through its attachment to the neck, enabling the detection of the degree of flexion. As demonstrated in Figure 7b, the (R − R_0_)/R_0_ ratio exhibited a substantially divergent response at 30° and 90° of neck flexion, while the signal pattern remained consistent across all angles. This outcome validates the sensor’s angle-independent stability. Of particular note is the observation that the 90° bend produced proportionally higher signal amplitudes, a finding that lends itself to the ability to perform quantifiable posture tracking. This feature has the potential to be incorporated into a health monitoring system, whereby a preset threshold is exceeded, triggering a posture adjustment alarm. The efficacy of the spinal motion monitoring function was evaluated through the attachment of the sensor to the lumbar region. Participants were requested to adopt an upright sitting posture, followed by flexion of the spine, and then a return to an upright sitting posture. Figure 7c demonstrates a discernible peak in bending time, which is in contrast to the signal consistent with the baseline in the upright sitting position. This disparity can be attributed to the fact that different poses result in different levels of stretching and bending of the sensor, leading to reorganization of the network inside the flexible sensor and the subsequent output of different electrical signals. These findings confirm the sensor’s ability to detect biomechanical anomalies and translate them into viable feedback information with great potential for habit correction and preventive healthcare applications.

### 3.8. Application of AgNW-TPU Sensor Arrays

In the domain of human–machine interaction, the objective is to facilitate the more natural and intelligent deployment of robots on a wider scale. To this end, our research was expanded to encompass the development of a 3 × 3 array sensor. As illustrated in Figure 1, the array sensor maintains a “sandwich” configuration, with the salient difference being that the macro structure of the inner layer network consists of 3 × 3 connection strips, denoted with A, B, and C and 1, 2, and 3, respectively, in both the horizontal and vertical directions (Figure 8). When the sensor is subjected to an external force, the horizontal 1, 2, and 3 and vertical A, B, and C points experience pressure; however, the pressure at the intersection points is the greatest. Consequently, the increased density of the array corresponds to a greater number of stress concentration points and more substantial feedback information from the robotic arm. This sensor is capable of determining an object’s weight and contact shape by assessing its pressure distribution characteristics. When the weight of the dumbbell is 10 g, the ΔR/R_0_ concentrated at point (1, A) is 7.67%, while stress at other points exists but is not zero. The signal difference between the intersection points and non-intersection points exceeds 5.1 times. When a doll with a wider base is used, the pressure at points (2, B), (2, C), (3, B), and (3, C) is the highest, with a matching degree of 85% with the doll’s base contact contour (Figure 8e). Therefore, when this sensor is attached to a glove to simulate a robot’s grasping motion, it can be observed that, compared to a single sensor that can only feedback total stress, its force control accuracy is better, providing more information and further proving its feasibility in human–machine interactions (Figure 8g).

### 3.9. Field of Human–Machine Interaction

AgNW-TPU flexible sensors have great advantages in human–computer interaction, where the flexible thin-film sensors can be attached to the robotic arm for swinging, and it can be seen that the bending and recovery phases show two peaks (Figure 9d). This may be due to the reduction in the conductive pathways within the sensor during bending, resulting in an increase in the relative resistance of the sensor. As the elbow joint of the robot arm recovers, the resistance of the sensor also recovers. Flexible sensors were attached to the human knuckle for 45° and 90° bending for mechanical finger simulation, which produced two electrical signals of different heights, reflecting good stability and fast response (Figure 9b). The flexible sensors were also attached to human wrist joints for both grasping and swinging actions, which also demonstrated their detectability and sensitivity (Figure 9a). To ensure the rigor of the experiment, we attached the sensor to the arm of the mechanical puppet and let the mechanical puppet perform such simple actions as rock, paper, and scissors using the manipulation command; it can be seen from Figure 9c that the sensor can still output stable and recognizable electrical signals to the machine. A team of volunteers was recruited to interpret the observed phenomena by solely monitoring the electrical signals. Subsequent to the repetition of the experimental procedure, the results indicated that the sensor demonstrated an electrical signal error of less than 1%.

Therefore, flexible AgNW-TPU sensors have a great advantage in human–computer interaction. In addition, the flexible sensor can output electrical signals through the tiny movement of the masseter muscle when it is attached to the face (Figure 9e). For example, when a volunteer says “Zhejiang Sci-tech University”, the electrical signal quickly outputs a series of folded lines, indicating that the flexible sensor can also be used for applications involving aphasia patients or crypto-language, among others.

## 4. Conclusions

In this study, the sensors were constructed with thermoplastic polyurethane (TPU) as the substrate and AgNW as the conductive layer, arranged in a “sandwich” structure. TPU films were fabricated using the electrospinning method. AgNWs with an aspect ratio of up to 1921 were prepared via the alcohol reduction method. An AgNW-TPU flexible thin-film sensor was subsequently developed using the drop-casting technique. The sensor exhibited a low initial resistance (26.7 Ω), a high strain range (0–150%), high sensitivity (GF up to 19.29), a fast response and recovery time (112 ms), excellent cycling durability (more than 3000 cycles), and excellent conductivity (428 S/cm). This sensor can be integrated with textiles for monitoring human health and motion as well as applied in human–computer interaction and encrypted communication scenarios. This work provides new directions for the application of AgNW-TPU flexible thin-film sensors.

## Figures and Tables

**Figure 1 sensors-25-05191-f001:**
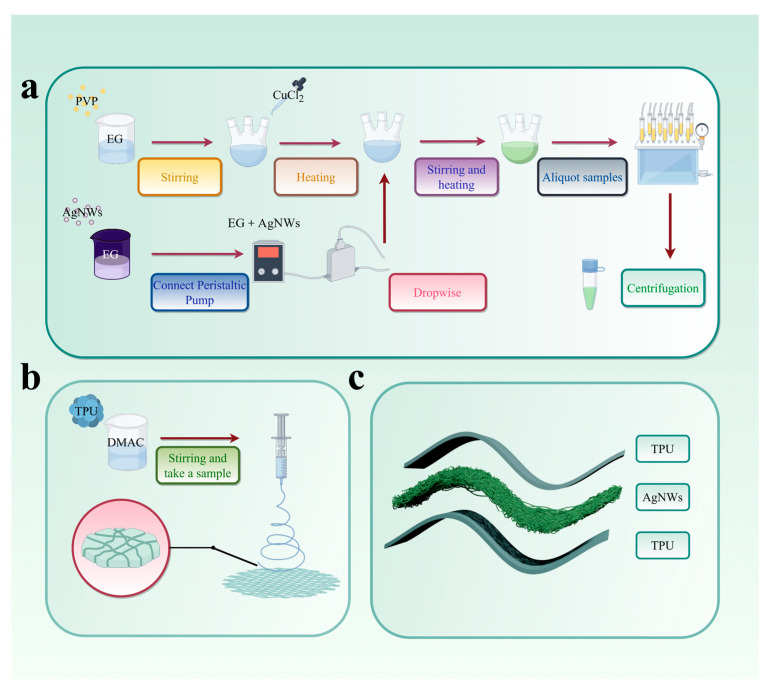
The production process of AgNWs (**a**). The production process of TPU-based nanofiber membranes (**b**). Architecture of the AgNW-TPU flexible film sensors (**c**) (by Figdraw, accessed on 10 June 2025).

**Figure 2 sensors-25-05191-f002:**
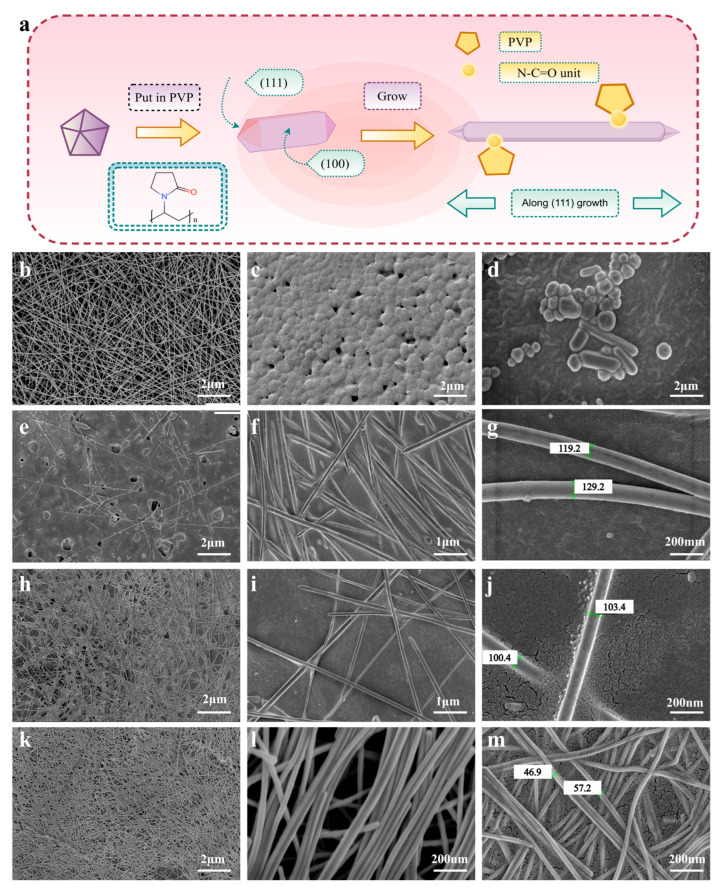
Schematic illustration of the growth of AgNWs (**a**) (by Figdraw, accessed on 10 June 2025); SEM images of (**b**) TPU-based nanofiber; (**c**) TPU made by the casting method; A6 (**e**–**g**), A7 (**h**–**j**), A8_1_ (**d**), and A8_2_ (**k**–**m**).

**Figure 3 sensors-25-05191-f003:**
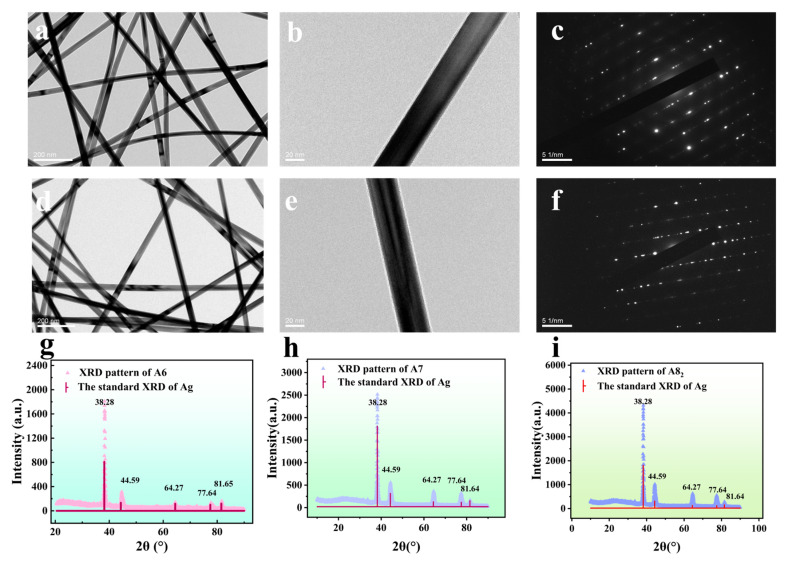
The surface state of AgNWs and the electron diffraction patterns composed of two-dimensional lattices (**a**,**b**,**d**,**e**), as photographed by a transmission electron microscope (**c**,**f**). XRD of A6 (**g**), A7 (**h**), and A8_2_ (**i**).

**Figure 4 sensors-25-05191-f004:**
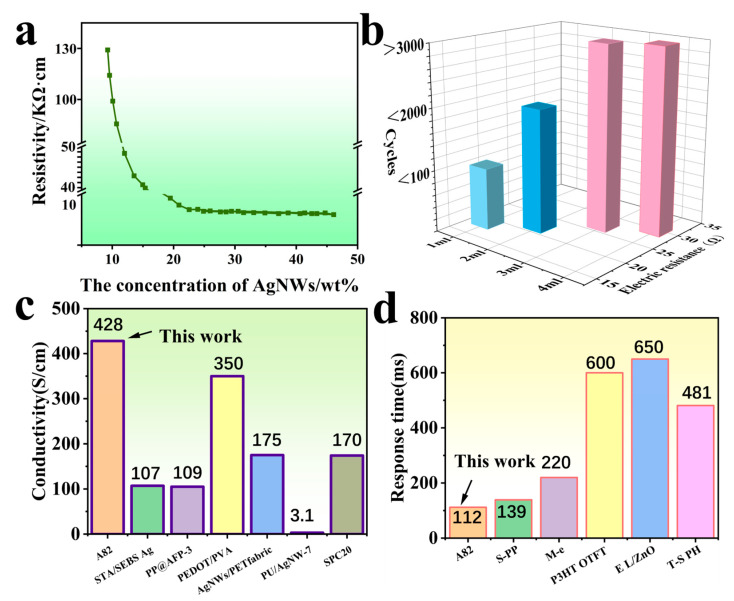
AgNWs’ concentration versus resistivity curves (**a**), resistance and number of cycles plotted for sensors at different doses (**b**), comparison of conductivity with this work (STA/SEBS Ag [32], PP@AFP-3 [33], PEDOT/PVA [34], AgNW/PET fabric [35], PU/AgNW-7 [36], and SPC20 [37]) (**c**), comparison of response times with this work (S-PP [38], M-e [39], P3HT OTFT [40], E L/ZnO [41], and T-S PH [42]) (**d**).

**Figure 5 sensors-25-05191-f005:**
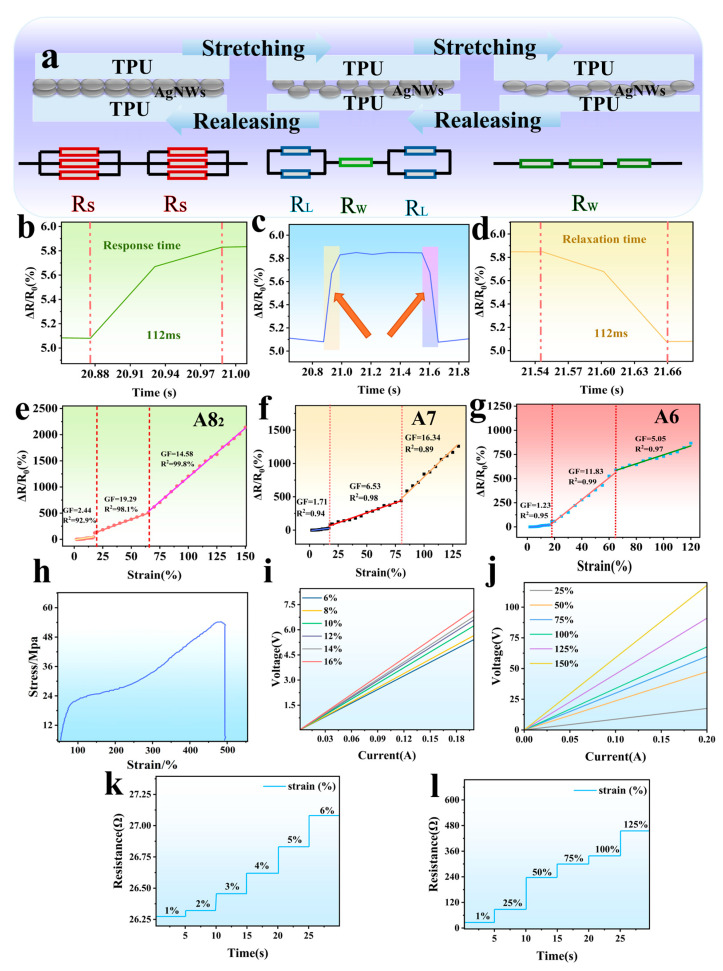
Schematic diagram of microscopic changes in sensor tensile resistance (**a**). Sensor response time curve (**b**–**d**). Tensile–resistance change rate curve of AgNW-TPU sensor response: A8_2_ (**e**), A7 (**f**), and A6 (**g**). Stress–strain curve of the sensor (**h**). The V-I curves under 0%~16% (**i**) and 25%~150% tensile strain (**j**). Dynamic tensile curve of the sensor from 0% to 5% (**k**). Dynamic tensile curve from 0% to 125% (**l**).

**Figure 6 sensors-25-05191-f006:**
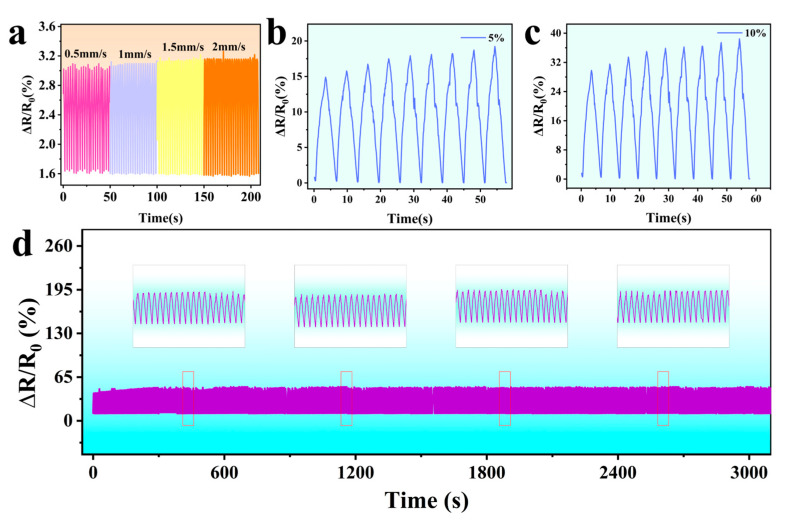
Cyclic resistance curves at the same strain (50%), and different speeds (0.5, 1.0, 1.5, and 2.0 mm/s) (**a**). Long-term strain sensing stability under 0–10% strain (**b**–**d**).

**Figure 7 sensors-25-05191-f007:**
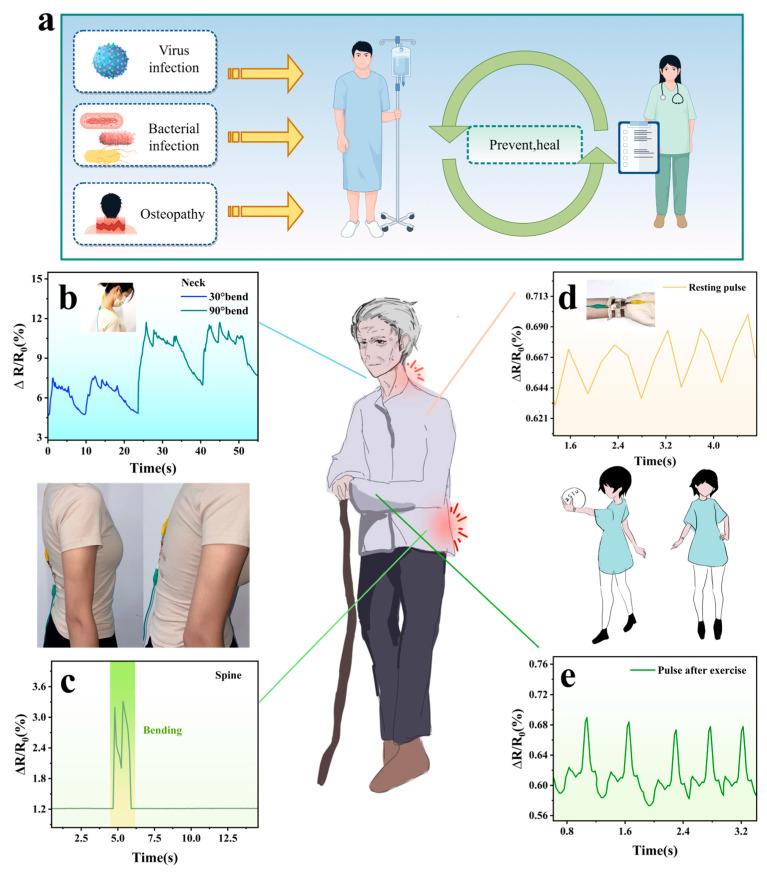
Flexible sensors can play a role in disease prevention (**a**) (by Figdraw, accessed on 10 June 2025). Electrical signals generated when the human neck bends (**b**). The electrical signals outputted due to the curvature of the spine (**c**). Flexible smart wristband for real-time monitoring of human pulse signal (**d**,**e**).

**Figure 8 sensors-25-05191-f008:**
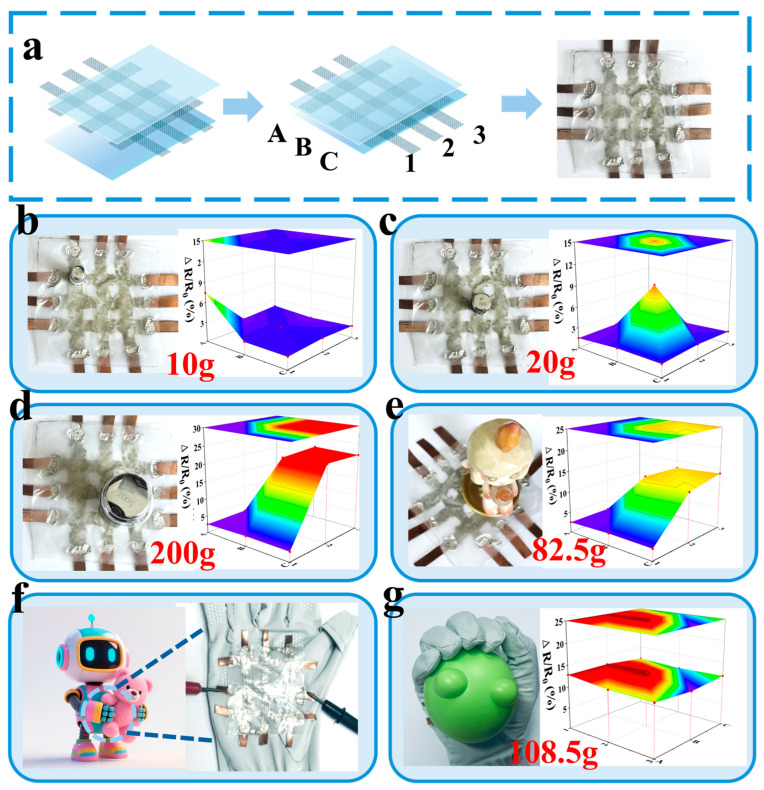
Schematic and physical drawings of the array sensors (**a**); electrical signals reflected on the array by different weights: 10 g (**b**), 20 g (**c**), 200 g (**d**); electrical signals generated by pressure from the dolls (**e**); sewing the sensors to the gloves to simulate movement (**f**); electrical signals generated by grasping the objects (**g**).

**Figure 9 sensors-25-05191-f009:**
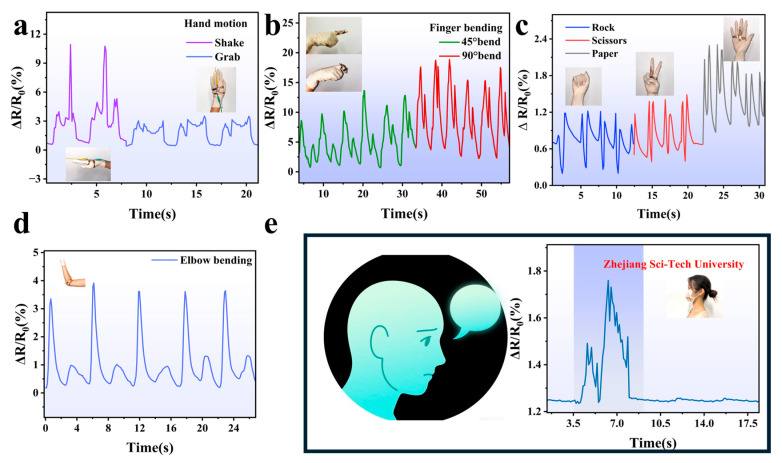
Electrical signals from simulated mechanical grasping and swinging motions (**a**). Finger bends at 45° and 90° (**b**). Robot plays rock, paper, and scissors (**c**). Elbow bending (**d**). Recognizing language through the movement of the masseter muscle, which can be used in healthcare (**e**).

## Data Availability

The data belongs to an ongoing research project. Due to data storage issues, the data presented in this article is not readily available at this time.

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
