# Peer review of "Preparation and Applications of Silver Nanowire-Polyurethane Flexible Sensor"

_sensors, 2025, doi:10.3390/s25165191_

Round 1

Reviewer 1 Report

Comments and Suggestions for Authors

The manuscript" Preparation and Applications of Silver Nanowire-Polyurethane Flexible Sensor" shows formation of sandwich structure of AgNWs spray coating on electrospinning TPU. In general using such combination for stretch sensors are not novel, done in various combination (https://doi.org/10.3390/nano12111932, https://doi.org/10.1039/C8TA11435H). The authors should state the novelty in this work. Additionally with only 25 references those are too few for such already widely published idea of stretching sensors. There are some main parts that needs revision.

  1. The main drawback which lowers the quality of this work are the missing discussion in the result part with so good as no reference to other work is provided. Also the differences between the reaction time as names A6-A8 and A81 and A82 are not so different as example if compared at Figure 3 g -i. The XRD peaks at same position only vary little with intensities. Also the SEM images did not show a clear separation of those. Please give more information what is different between those samples. Hence, no information is given how many samples are studied, the small differences can be also laying in the standard deviation if several samples would be used.
  2.  The main part of the result is dedicated to the sensor function. It would be also important to add the elastic young's modulus of the sandwich structure and there dependency on stretch how the resistivity change. If possible add those.
  3. The scheme 1 a shows some action said dropwise. For which does that count and what was added. Additionally at scheme 1b the sandwich structure are more or less shown that AgNWs are a separate unit but those are coated on the TPU. In which thickness are those AgNWs coatings found and what is the optimal thickness related to resistivity.
  4. The good results of this works are difficult to compare to other works in the field of strain sensors. As there is no discussion provided authors need to include a table of comparison regarding strain response, conductivity (please show S/cm) and GF on end of the result part. There are several different concepts which in general should be compared at least in the introduction. Some authors used composite TPU AgNWs (https://doi.org/10.1021/acsami.9b08611) showing higher performance and accuracy in strain sensors. What advantage bring the authors concept as if said such used in healthcare, probably healthcare vest with those need have functionality after washing might not be given with authors design. Please create some discussion of such.

Author Response

请参阅附件。

Reviewer 2 Report

Comments and Suggestions for Authors

This manuscript reports a flexible sensor based on AgNWs and electrospun TPU with good sensitivity and mechanical stability. The topic is relevant to the Sensors journal, and the results are promising. However, several important issues must be addressed before acceptance:

  1. The novelty of the sensor design is not clearly distinguished from existing AgNW-TPU structures. Authors should clarify what is new.
  2. Performance comparison with other similar sensors is missing. A benchmarking table is needed.
  3. The sensing mechanism explanation lacks depth and scientific basis.
  4. Application tests are shown but lack quantitative evaluation, especially in real-use scenarios.
  5. English grammar and figure clarity should be improved throughout.

Therefore, I recommend accepting this manuscript for publication in this journal with major revision.

Author Response

请参阅附件

Reviewer 3 Report

Comments and Suggestions for Authors

The article titled “Preparation and Applications of Silver Nanowire-Polyurethane Flexible Sensor” by JiangYin Shan, et, al. reported a "sandwich" structured flexible sensor exhibit a high strain range and high sensitivity. The data analysis is relatively comprehensive; however, it lacks clear innovation and a focused direction. The overall organization should be improved by selecting and emphasizing the most important data, rather than presenting all the experimental results without prioritization. Unfortunately, after careful review, I believe the manuscript does not currently meet the publication standards required by the journal.

The main concerns include:  

  1. The GF is not over 1703 during 0%-150%; please correct this.
  2. In Section 3.1, the detailed mechanism behind the decrease in AgNW diameter and the increase in crystallinity with prolonged reaction time should be further elaborated.
  3. In the XRD results, samples A6 and A7 exhibit the same diffraction peaks with only differences in intensity. An additional explanation or discussion of this observation is needed.
  4. In section 3.4, the author spends a long paragraph explaining the possible stretching sensing behavior from 0-150%; however, this lacks supporting evidence. Moreover, it is uncommon for a stretch sensor to exhibit four distinct sensing regions. It is recommended to perform repeated stretch sensing tests to validate this behavior.
  5. In Figure 4b, it is recommended to replace the current plot with a complete stretching and release cycle, including both response time and recovery time, as these are critical performance parameters.
  6. In Figure 5a, the resistance change shows a significant increase over time.
  7. Figure 6 overlaps with Figures 7 and 8 in terms of content. The motion signals for jumping, running, walking, and squatting appear very similar in both pattern and intensity. It is suggested to remove it.
  8. For all real demonstration images, the sensor attached to the body should be shown.
  9. There are some spelling and labeling issues: “the patterns. see that”, “Figure 4(a)”, etc.

Author Response

请参阅附件。

Round 2

Reviewer 1 Report

Comments and Suggestions for Authors

The authors revised the manuscript and gave good response to each question and suggestion. No further comments.

Author Response

感谢您抽出宝贵时间审阅我们的文章,您的宝贵意见使修改后的稿件具有可读性和科学性,感谢您的付出和支持。

Reviewer 2 Report

Comments and Suggestions for Authors

Through the kind and adequate responses of authors, the doubts are dispelled and the errors of the manuscript are correctly revised. I think the authors provide quite reasonable changes.

Thus, I gladly recommend that this manuscript might be suitable to publish to this journal.

Author Response

Thank you for taking your valuable time to review our article, your valuable comments make the revised manuscript readable and scientific enhancement, thank you for your dedication and support.

Reviewer 3 Report

Comments and Suggestions for Authors

The author has addressed most of my previous concerns in detail, and the paper has improved a lot. There remain a few minor issues that are suggested for revision:

In line 153, The manuscript states that “the diameter of A82 (~80 nm) is significantly smaller than that of A7 (~100 nm) and A6 (~120 nm).” It is recommended to provide high-magnification SEM images for direct visual comparison of the nanowire diameters.

While the synthesis section includes comprehensive characterization, to further emphasize the advantage of high-aspect-ratio AgNWs, it is recommended to establish a clearer link between these material variations and final device performance. Specifically, comparing the sensing performance using AgNWs with different aspect ratios (e.g., in Fig. 5e) would strengthen this connection.

There appears to be a gap in the data between 10–20% in Fig. 4a. Please clarify it.

It is suggested to include a real photograph showing the sensor in use for pulse detection inf fig 7d and e, to better illustrate the experimental setup.

Some minor issue: “the reaction solution solution of AgNWs”, figure 5 (a)(b、c、d), etc.
